# Technical note: Lowermost-stratospheric moist bias in ECMWF IFS model diagnosed from airborne GLORIA observations during winter/spring 2016

Wolfgang Woiwode[1], Andreas Dörnbrack[2], Inna Polichtchouk[3], Sören Johansson[1], Ben Harvey[4], Michael Höpfner[1], Jörn Ungermann[5], and Felix Friedl-Vallon[1]

[1]Institute of Meteorology and Climate Research (IMK), Karlsruhe Institute of Technology (KIT), Karlsruhe, Germany
[2]Deutsches Zentrum für Luft- und Raumfahrt, Institut für Physik der Atmosphäre, Oberpfaffenhofen, Germany
[3]European Centre for Medium-Range Weather Forecasts, Reading, UK
[4]National Centre for Atmospheric Science, University of Reading, Reading, United Kingdom
[5]Institute of Energy and Climate Research – Stratosphere (IEK-7), Forschungszentrum Jülich, Jülich, Germany

*Correspondence to*: Wolfgang Woiwode (wolfgang.woiwode@kit.edu)

**Abstract.** Numerical weather forecast systems like the ECMWF IFS (European Centre for Medium-Range Weather Forecasts – Integrated Forecasting System) are known to be affected by a moist bias in the extratropical lowermost stratosphere (LMS) which results in a systematic cold bias there. We use high spatial resolution water vapour measurements by the airborne infrared limb-imager GLORIA (Gimballed Limb Observer for Radiance Imaging of the Atmosphere) during the PGS (POLSTRACC/GW-LCYLCE-II/SALSA) campaign to study the LMS moist bias in ECMWF analyses and 12 h forecasts from January to March 2016. Thereby, we exploit the 2-dimensional observational capabilities of GLORIA, when compared to in situ observations, and the higher vertical and horizontal resolution, when compared to satellite observations. Using GLORIA observations taken during five flights in the polar sub-vortex region around Scandinavia and Greenland, we diagnose a systematic moist bias in the LMS exceeding +50 % (January) to +30 % (March) at potential vorticity levels from 10 PVU (~highest level accessed with suitable coverage) to 7 PVU. In the diagnosed time period, the moist bias reduces at the highest and driest air masses observed, but clearly persists at lower levels until mid-March. Sensitivity experiments with more frequent temporal output, and lower/higher horizontal and vertical resolution, show the short-term forecasts to be practically insensitive to these parameters on time scales of < 12 hours. Our results confirm that the diagnosed moist bias is present already in the initial conditions (i.e., the analysis) and thus support the hypothesis that the cold bias develops as a result of forecast initialisation. The moist bias in the analysis might be explained by a model bias together with the lack of water vapour observations suitable for assimilation above the tropopause.

## 1 Introduction

Accurate representation of water vapour in the lowermost stratosphere (LMS) is important for numerical weather forecasting and climate simulations. Water vapour mixing ratios in the tropopause region affect the temperature distribution and the location of the thermal tropopause, and hence stratospheric dynamics (Stenke et al., 2008 and references therein). Radiative forcing has been shown to respond sensitively to changes in LMS water vapour mixing ratios (Solomon et al., 2010; Riese et al., 2012). Furthermore, water vapour in the tropopause region controls the formation of high-altitude cirrus clouds and contrails.

Atmospheric general circulation models are known to be affected by a systematic cold bias in the extratropical LMS, which is strongest in the summer hemisphere, but significant also in the winter hemisphere (Gates et al., 1999; Stenke et al. 2008 and references therein). The cold bias is known to be the consequence of a moist bias which results in too strong longwave cooling. State-of-the-art high-resolution numerical weather prediction systems such as the ECMWF IFS (European Centre for Medium-Range Weather Forecasts – Integrated Forecast System) are also affected by this cold bias (Hogan et al., 2017; Shepherd et al., 2018) at all forecast ranges and at all resolutions. As specific humidity observations are not assimilated above the

tropopause, accurate observations of water vapour in the LMS with wide coverage and high spatial resolution are required to validate analyses and forecasts and to aid in model development.

Airborne remote sensing observations using lidar or infrared limb sounding fill the observational "gap" between focused in situ and global satellite observations in terms of spatial coverage and resolution. They allow to study mesoscale water vapour distributions across the tropopause with high vertical and horizontal resolution (e.g. Flentje et al., 2005; Ungermann et al., 2012; Schäfler et al., 2018; Woiwode et al., 2018). Here, we use observations by the infrared limb imager GLORIA (Gimballed Limb Observer for Radiance Imaging of the Atmosphere) (Friedl-Vallon et al., 2014; Riese et al., 2014) to quantify the LMS moist bias under Arctic winter and spring conditions. In particular, we investigate the development of the moist bias from January to March 2016. For a flight on 26 February 2016, we furthermore discuss moist bias sensitivity in short 12 h forecasts with more frequent temporal output, and lower/higher horizontal and vertical resolutions. In Section 2, we introduce the data and diagnostics used. The results are presented in Section 3, and discussed in Section 4. Conclusions are given in Section 5.

## 2 Data and Methods

### 2.1 GLORIA observations in Arctic winter 2015/16

GLORIA is an airborne thermal infrared limb-imaging Fourier transform spectrometer (Friedl-Vallon et al., 2014). It was deployed onboard the German High Altitude and Long Range Research Aircraft (HALO) during the combined PGS (POLSTRACC/GW-LCYCLE II/SALSA) field campaign in the Arctic winter 2015/16 (Oelhaf et al., 2019). The PGS campaign was designated to study the polar stratosphere in a changing climate, the life cycle of gravity waves, and the seasonality of air mass transport and composition in the LMS. Based in Oberpfaffenhofen (Germany) and Kiruna (Sweden), HALO enabled maximum flight distances exceeding ~8000 km and ceiling altitudes exceeding ~14 km.

GLORIA measures infrared spectra in the spectral range from 780 to 1400 $cm^{-1}$ and views to the right hand side of the flight track. From the spectra, vertical profiles of temperature, trace gases, and cloud parameters are retrieved. Here, we use GLORIA observations in the high spectral resolution "chemistry mode" which involves a spectral sampling of 0.0625 $cm^{-1}$ and an associated horizontal sampling of ~3 km. The water vapour data is characterized by a typical vertical resolution of 400-700 m and combined random and systematic 1σ errors of typically 10-20 % (Johansson et al., 2018). The errors are expected to cancel out mostly when the data are analysed as ensemble (e.g. in correlation analyses). As analyzed by Johansson et al. (2018), the median difference and the median absolute deviation between GLORIA and FISH (Fast In situ Stratospheric Hygrometer, Zöger et al. 1999; Meyer et al., 2015) in situ water vapour observations in the Upper Troposphere/Lowermost Stratosphere (UT/LMS) during PGS are only 0.13 ppmv and ±0.63 ppmv, respectively. From the water vapour profiles derived from the GLORIA observations, 2-dimensional vertical cross sections of water vapour along the HALO flight tracks are constructed.

In the present study, we use GLORIA observations during five Arctic flights. The flights on 12 January 2016, 18 January 2016, and 20 January 2016 provide a robust estimate of the LMS moist bias in mid-winter, since extended 2-dimensional water vapour distributions associated with independent flights and different meteorological scenarios are analysed. The flights on 26 February 2016 and 13 March 2016 allow us to investigate how the moist bias develops in the late winter and early spring. The choice of the shown data was constrained by the dates of the flights, availability of the GLORIA "chemistry mode" data, observations under sufficiently cloud-free conditions, and availability of observations within the LMS in the polar sub-vortex region (for explanation of "sub-vortex" region, see e.g. Werner et al., 2010). In the following, we show 2-dimensional vertical cross-sections of GLORIA water vapour observations along the HALO flight tracks and compare the observations to the ECMWF system.

## 2.2 ECMWF IFS data

The ECMWF IFS is a global weather forecasting and analysis system (https://www.ecmwf.int/en/research/modelling-and-prediction) based on a semi-Lagrangian hydrostatic formulation. Between 26 June 2013 and 8 March 2016, the high-resolution forecasts and analysis were at $T_L1279L137$ resolution, corresponding to 16 km in the horizontal and 300-400 m in the vertical at the tropopause. On 8 March 2016, the horizontal resolution was upgraded to 9 km (or $T_{Co}1279$) (Hólm et al., 2016; Malardel and Wedi, 2016), which was made possible by the introduction of a cubic octahedral ($T_{Co}$) grid (Malardel et al., 2016; see also Wedi, 2014). For comparison to GLORIA observations, we use 00 UTC and 12 UTC analysis and hourly output from a 12 h deterministic forecasts in between the analysis cycles.

In particular, we compare the forecasted specific humidity ($q_v$), converted to volume mixing ratio (parts per million by volume, ppmv), with gas-phase water vapour volume mixing ratios derived from the GLORIA observations. The model output is interpolated in space and time to the geolocations of the tangent points of the GLORIA limb observations. In this manner, vertical cross sections of IFS water vapour corresponding to the vertical cross sections derived from the GLORIA observations are obtained. For vertical assignment of comparable air masses during the winter, we use potential vorticity (PV, unit: PVU) interpolated from the IFS in the same way.

We furthermore perform short (< 12 h) sensitivity forecasts with higher frequency of temporal output (450 s instead of 1 h), lower/higher horizontal resolution ($T_{Co}319$ or ~36 km; $T_{Co}639$ or ~18 km; and $T_{Co}1279$ or ~9 km all on cubic octahedral grid; instead of the operational $T_L1279$ or ~18 km resolution on the linear grid) and lower/higher vertical resolution (91 and 198 levels instead of 137 levels) to investigate whether the moist bias is sensitive to these model changes. These sensitivity forecasts are performed for comparison with the 26 February 2016 flight and have all been initialized from the operational 4D-Var analysis with the outer loop trajectory performed at $T_L1279L137$ resolution (and the three inner loops are performed at $T_L399L137$, $T_L319L137$ and $T_L255L137$ resolution).

## 2.3 Data selection and correlations analysis

The first step in our analysis is the identification of flight sections located in the LMS and inside the polar sub-vortex region. To identify sub-vortex air masses, we analyse vertical cross-sections of water vapour retrieved from the GLORIA observations and interpolated from the IFS in combination with potential vorticity interpolated from the IFS (Fig. 1). Air masses located in the sub-vortex LMS are characterized by low humidity and a low tropopause. We use the 2 PVU level as an indicator for the dynamical tropopause.

Using these parameters, the LMS in the sub-vortex region can be clearly identified in the vertical cross sections and the PV maps, as shown in Figures 1a-d for the flight on 12 January 2016. To quantify the bias, we use flight sections with the dynamical tropopause being mostly located below 10 km. Regions characterised by strong horizontal gradients are avoided, since certain features may be forecasted in a realistic way but do not exactly coincide with the observed location, thus inducing an overestimation of differences between forecast and observation (compare Fig 1a-c tropopause fold between 11:00 and 12:00 UTC). The section of the discussed flight used in our analysis is marked by blue dashed arrows in Figures 1a and 1d.

The residuals between the vertical cross sections show the moist bias of the ECMWF IFS data relative to the GLORIA observations. For the flight on 12 January 2016, the LMS moist bias can be clearly identified above the tropopause, in particular in the sub-vortex region after 12 UTC (Fig. 1c). To quantify the moist bias in the sub-vortex region, we correlate IFS specific humidity with specific humidity measured by GLORIA in the selected flight sections (see Fig. 1e). Vertical assignment of the selected sub-vortex data points is done using the IFS PV data. Using this quantity, dynamically similar air masses can be compared during the course of the winter, which is not possible using for example geometric altitude or potential temperature due to diabatic air mass descent. Furthermore, the mean correlation of the selected data points is shown for quantification (Fig. 1e). Finally, we calculate the mean bias and the standard deviation of the individual data points at selected, rounded levels of potential vorticity. Note that slightly different values are diagnosed here when compared to the mean correlations, since the

mean correlations shown in the correlation plots are a function of volume mixing ratio and not potential vorticity. However, the overall conclusions are the same in both cases.

## 3 Results

In Figure 1a-c, we present the GLORIA and IFS data corresponding to the flight on 12 January 2016 and their residuals. During this flight, a wide range of sub-vortex air masses characterized by high PV values (Fig. 1d) was accessed from the Alps to the Arctic Sea after crossing the polar front jet stream. A developed tropopause fold is clearly and consistently identified in both observations and forecast between 11 and 12 UTC (see also Woiwode et al. 2018). North of the tropopause fold, a lower tropopause and a mostly unperturbed LMS is found in the polar sub-vortex region.

In the residual, noticeable differences between observation and ECMWF analysis and forecasts are found (Fig. 1c). In the first flight part before 12 UTC, positive and negative residuals are mostly a consequence of differences inside the tropopause fold and further mesoscale fine-structures. North of the tropopause fold, a relatively homogeneous systematic moist bias is clearly identified. Figure 1e shows the correlation of the selected IFS data with the GLORIA observations. While the whole ensemble of data points (grey) is mostly scattered around the 1:1 line (yellow solid line), the colour-coded data points beyond 12 UTC

(see blue arrows in Fig. 1a,d) clearly show the moist bias increasing with PV. The observed average bias slightly exceeds +50 % at 9 to 10 PVU in Fig. 1e. When the data points are filtered for a rounded potential vorticity of 10 PVU, a mean bias of +70 % and a standard deviation of 15 % are diagnosed at the highest and driest levels that were accessed. Note that individual data points scattering to very dry values in the correlation (see also corresponding panels in Fig. 2) are a consequence of scattering of the GLORIA data within their random and systematic uncertainties (see Sect. 2.1) and are not indicative of e.g.

extreme dehydration events. Such data points are furthermore emphasized due to the large overall number of data points (i.e. overlapping of data points) and by the logarithmic scale of the x-axes.

  Using the same approach, the subsequent flights are analysed in Figure 2. The GLORIA observations of the flight on 18 January 2016 (Fig. 2a) were performed in a partly perturbed sub-vortex region, with structures characterised by lower PV stretching into the air volume observed by GLORIA (Fig. 2b). The correlation of the IFS and GLORIA data shows a systematic

moist bias of the IFS data, which is lower than during the previous flight. When the data points are filtered for a rounded potential vorticity of 9 PVU, a mean bias of +32 % and a standard deviation of 25 % are diagnosed. The observed decreasing mean bias towards highest PV values is attributed to the structures from outside the sub-vortex region which are not affected by the moist bias. For the flight on 20 January 2016, again a slightly lower systematic moist bias is found in the IFS data when compared to the flight on 12 January 2016. When the data points are filtered for a rounded potential vorticity of 9 PVU, a

mean bias of +45 % and a standard deviation of 19 % are diagnosed.

  Note that in situ comparisons with FISH show water vapour mixing ratios measured by GLORIA during PGS until end of January 2016 to be systematically lower by 0.01 ppmv to 0.75 ppmv at flight altitude (Johansson et al., 2018). Differences between FISH and GLORIA practically cancel out on average in February and March. While the results in January might be partially caused by the different sampling characteristics of the GLORIA limb observations when compared to in situ

observations (e.g. GLORIA viewing deeper into sub-vortex air masses in some cases), remaining issues in the calibration of GLORIA data version used here cannot be excluded. To avoid a potential overestimation of the moist bias peak value in the IFS data in January, we therefore provide a conservative estimate for the flight on 12 January 2016 of > +50% at 10 PVU instead of the value of +70% (see above) to account for this potential uncertainty.

  At the end of the winter, during the flight on 26 February 2016, largely unperturbed sub-vortex air masses were probed by

GLORIA from East Canada to West Greenland (Fig. 2g, h). For the correlation analysis, we use the data points characterised by strongest downwelling (dashed blue arrows in Fig. 2g, h). Here, the mean moist bias peaks at 7 PVU, stretches down to ~4

PVU, and decreases also towards higher PV values (Fig. 2i). When the data points are filtered for a rounded potential vorticity of 7 PVU (10 PVU), a mean bias of +38 % (+14 %) and a standard deviation of 16 % (14 %) are diagnosed.

During the first flight of the double flight on 13 March 2019 (see Oelhaf et al., 2019; only first flight used in Fig. 2j-l), again largely unperturbed sub-vortex air was probed in early spring. Similar to the previous flight, the average moist bias peaks at lower altitudes. When the data points are filtered for a rounded potential vorticity of 7 PVU (10 PVU), a mean bias of +35 % (+6 %) and a standard deviation of 23 % (11 %) are diagnosed. Recall that the horizontal resolution of the operational high resolution ECMWF system was upgraded from 18 km to 9 km on 8 March 2019. Therefore, the presence of the moist bias for the 13 March 2019 flight indicates that the bias is unaffected by this horizontal resolution upgrade. In all correlation analyses, the average correlation (cyan solid line) is mostly situated close to the 1:1 line around and below the dynamical tropopause.

In Figure 3, an overlay of the mean IFS/GLORIA correlations is shown for all flights except of the flight on 18 January 2016, which is excluded due to the effects of air masses from outside the sub-vortex region (see above). The overlay shows that the moist bias of the IFS data is largest on 12 January 2016 and peaks in the highest and driest air masses accessed by the observations. During the subsequent flights, the mean bias systematically declines in the highest/driest air masses observed. In February and March, the mean bias persists and still approaches peak values exceeding 30 % at a potential vorticity of 7 PVU.

The mean IFS/GLORIA correlations for the 12 h forecast sensitivity experiments for the flight on 26 February 2016 including more frequent temporal output, and higher/lower horizontal and vertical resolutions are shown in Figure 4. None of the experiments notably affect the resulting mean correlation of the analysed short-term forecasts.

## 4 Discussion

Several possible reasons for the moist bias have been previously discussed in the literature. Due to a sharp water vapour contrasts present around the tropopause it is known that numerical diffusion (both explicit and implicit) can lead to too strong water vapour "leakage" from the moist troposphere into the dry stratosphere in low resolution model simulations. Mesoscale fine structures such as tropopause folding, intrusions and filamentary structures on horizontal and vertical scales smaller than the model resolution are likely to contribute to the observed moist bias. Consistently, moderate increase in model resolution (up to 60 km in the horizontal and up to 1 km in the vertical at the tropopause) has been shown to reduce the moist bias and consequently the cold bias (Roeckner et al., 2006, Polichtchouk et al., 2019). Nevertheless, ECMWF forecasts at 9-18 km horizontal resolution and better than 400 m vertical resolution at the tropopause are still affected by the cold bias in the mid-latitude and polar LMS (Shepherd et al., 2018). Our long-range 2-dimensional observations clearly confirm the moist bias to be present at such a high resolution and furthermore show how the moist bias develops from January to March in the vertical domain.

Since water vapour is not assimilated by ECMWF above the tropopause in any analysis/reanalysis products, it is possible that the cold bias in high resolution forecasts develops as a result of initialisation from too moist analysis in the mid-latitude and polar LMS. Indeed, using CARIBIC (Civil Aircraft for the Regular Investigation of the atmosphere Based on an Instrument Container) in situ observations onboard passenger aircraft from 2005-2012, Dyroff et al. (2015) found specific humidity in ECMWF analyses and 18 h and 24 h forecasts to be overestimated by 100 to 150% in the summer and autumn and by 50 to 100% in the winter and spring. They suggest that the observed moist bias is caused by small-scale stratospheric intrusions which are still unresolved by the model, numerical diffusion of water vapour across the hydropause from the advection scheme and a lacking constraint on humidity in the stratosphere. Other possible model processes contributing to the moistening of the LMS include vertical diffusion parametrization.

Consistently, Kaufmann et al. (2018) found a wet bias of 100% to 150% in the lowermost mid-latitude stratosphere in the ECMWF system in late spring 2014 and attributed it mainly to a too weak of a humidity gradient at the tropopause in the

model. Consistent with Kuntz et al. (2014), who analysed in situ observations from 2001 to 2011 and also diagnosed a wet lowermost stratosphere bias in the ECMWF system, Kaufmann et al. (2018) found the wet bias to decrease significantly towards higher altitudes.

It should be kept in mind that the previous studies covered years from 2001 to 2014, making direct comparisons to the more recent ECMWF system with better horizontal and vertical resolution difficult. However, while a mostly lower wet bias is diagnosed in our study, our results confirm the conclusion that the moist bias is already present in the ECMWF analyses during forecast initialization. Our results furthermore show that the bias is unaffected by the forecast resolution on short (< 12 h) time scales. Similar to the previous studies, we find the wet bias to decrease towards higher altitudes. Thereby, we characterize the analysed levels by potential vorticity rather than geometric altitude or potential temperature and find a coherent picture of the moist bias evolution from January to March 2016.

Using satellite observations, Hogan et al. (2017) and Shepherd et al. (2018) also found that specific humidity in the polar LMS is overestimated in the ECMWF analysis when compared to MLS (Microwave Limb Sounder observations). However, MLS observations have a comparably poor vertical resolution of ~ 5 km in the LMS, making the exact quantification of the moist bias across the sharp tropopause difficult. Therefore, global high vertical and horizontal resolution observations of water vapour across the tropopause, such as demonstrated by state-of-the-art airborne remote sensing instruments, would be desirable to better quantify and to help resolve the moist bias in the ECMWF system.

**5 Conclusions**

The comparison of state-of-the art high-resolution ECMWF analysis and short forecasts with high resolution GLORIA observations clearly shows a systematic moist bias in the ECMWF system peaking at 7 to 10 PVU. The moist bias reduces at the highest and driest levels at 8 to 10 PVU from mid-winter to early spring, but persists until mid-March at lower levels in the LMS. It extends down to altitudes below 8 km in strongly subsided air masses at the end of February. Sensitivity forecasts using more frequent temporal output, and higher/lower horizontal and vertical resolutions show practically no response of the mean bias to these changes. While it is possible that for longer lead times resolution will have an impact on the moist bias, we note that a unique 1 km horizontal resolution seasonal forecast with the IFS has a moister LMS than in a similar forecast at 9 km horizontal resolution (Wedi et al., 2020), implying that the lack of horizontal resolution is not the reason behind this bias in the forecasts. We also note that vertical resolution increase beyond 137 levels does not reduce the LMS cold bias in the medium range forecasts with IFS (Polichtchouk et al., 2019). It should be emphasized that all the sensitivity forecasts here were started from the same operational analysis. If the 4D-Var analysis itself was performed at different resolutions, the conclusion might be different. Similar to previous studies, the presented results support the conclusion that the moist bias is already present in ECMWF analysis – during forecast initialization. Our results show furthermore that on short (< 12 h) time scales the bias is unaffected by the forecast resolution.

The moist bias in the ECMWF analysis could be explained by the lack of observational constraint on specific humidity, as water vapour observations are not assimilated above the tropopause (i.e., the humidity increments are switched off above the hygropause, while temperature and winds are assimilated and thus affect moisture analysis). Therefore, the lower stratospheric moist bias in the analysis is dominated by errors in the model allowing water vapour leakages into the LMS. One possibility to minimize the LMS moist bias in the ECMWF system might be the systematic correction of the water vapour fields above the tropopause. This would, however, require a comprehensive characterisation of the moist bias in the extratropical LMS during the different seasons and a robust identification of the tropopause, also in dynamically perturbed regions. Thereby, data from further field campaigns like PGS, regular passenger aircraft observations such as CARIBIC, and the SPARC initiative (see https://www.sparc-climate.org/activities/water-vapour/) could be helpful. Another possibility could be the assimilation of

future space-borne global water vapour observations (e.g. infrared limb-imaging or lidar) with high spatial resolution in the LMS.

**Author contributions**

IP and BH conceived the study. WW, AD and IP elaborated the analyses. WW wrote the manuscript, with contributions from all co-authors. SJ, MH, JU and WW processed and analysed the GLORIA data, with further contributions by the GLORIA team from KIT and JÜLICH. FFV and the GLORIA team performed the GLORIA measurements and operations.

**Acknowledgements**

We acknowledge support by the German Research Foundation (Deutsche Forschungsgemeinschaft, DFG Priority Program SPP 1294) in general and, in particular, through research grant No WO 2160/1-1. We furthermore acknowledge partial support by BMBF within the research initiatives ROMIC (project GW-LCYCLE, subproject 2, 01LG1206B) and ROMIC II (project WASCLIM, subproject 5, 01LG1907E). We acknowledge ECMWF for providing the IFS data. We thank Elias Holm for helpful discussion. We thank the PGS coordination and flight planning teams, the GLORIA team from KIT and JÜLICH, and
DLR-FX for the planning and carrying out the flights and observations. We thank two anonymous Reviewers for valuable comments which helped us to improve the manuscript.

**Data availability**

The GLORIA observations can be accessed at the HALO database (https://doi.org/10.17616/R39Q0T, HALO consortium, last access: 16 April 2020) and at the KITopen repository (https://doi.org/10.5445/IR/1000086506, last access: 16 April 2020).
The IFS data are available via the ECMWF website (https://www.ecmwf.int/, ECMWF, last access: 16 April 2020). Information on SPARC (Stratosphere-troposphere Processes And their Role in Climate) water vapour assessments can be found at the SPARC website (https://www.sparc-climate.org/activities/water-vapour/, last access: 16 April 2020).

**Conflicts of interest**

The authors declare no conflicts of interests.

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

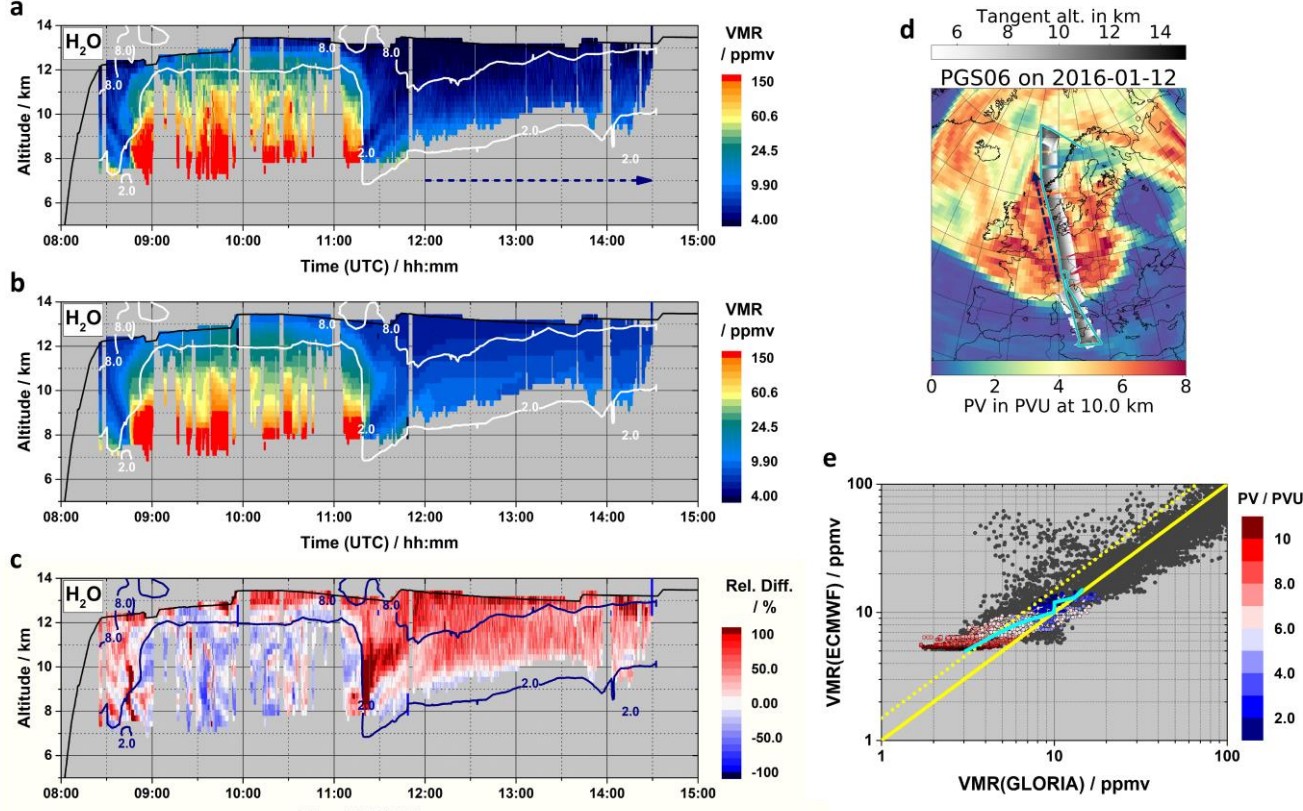

**Figure 1:** Principle of LMS moist bias quantification. Vertical cross-section of water vapour during the flight on 12 January 2016 (a) derived from GLORIA observations and (b) forecasted by the IFS. (c) Relative difference of water vapour IFS minus GLORIA. Solid contour lines (white and dark blue, respectively) in (a-c) indicate the PV levels of 2 and 8 PVU. Black solid lines indicate the flight altitude. (d) Flight path of HALO (cyan), tangent points of GLORIA limb observations (grey to white dots) and PV field at 10 km (contour). (e) Correlation between water vapour IFS and GLORIA (grey). Data points selected for the quantification are colour-coded with PV (see flight sections marked by dashed dark blue lines in (a,d)) and average line (cyan). The yellow solid line denotes a 1:1 correlation and the yellow dotted line a bias of +50%. Panels (a,b) modified from Woiwode et al. (2018).

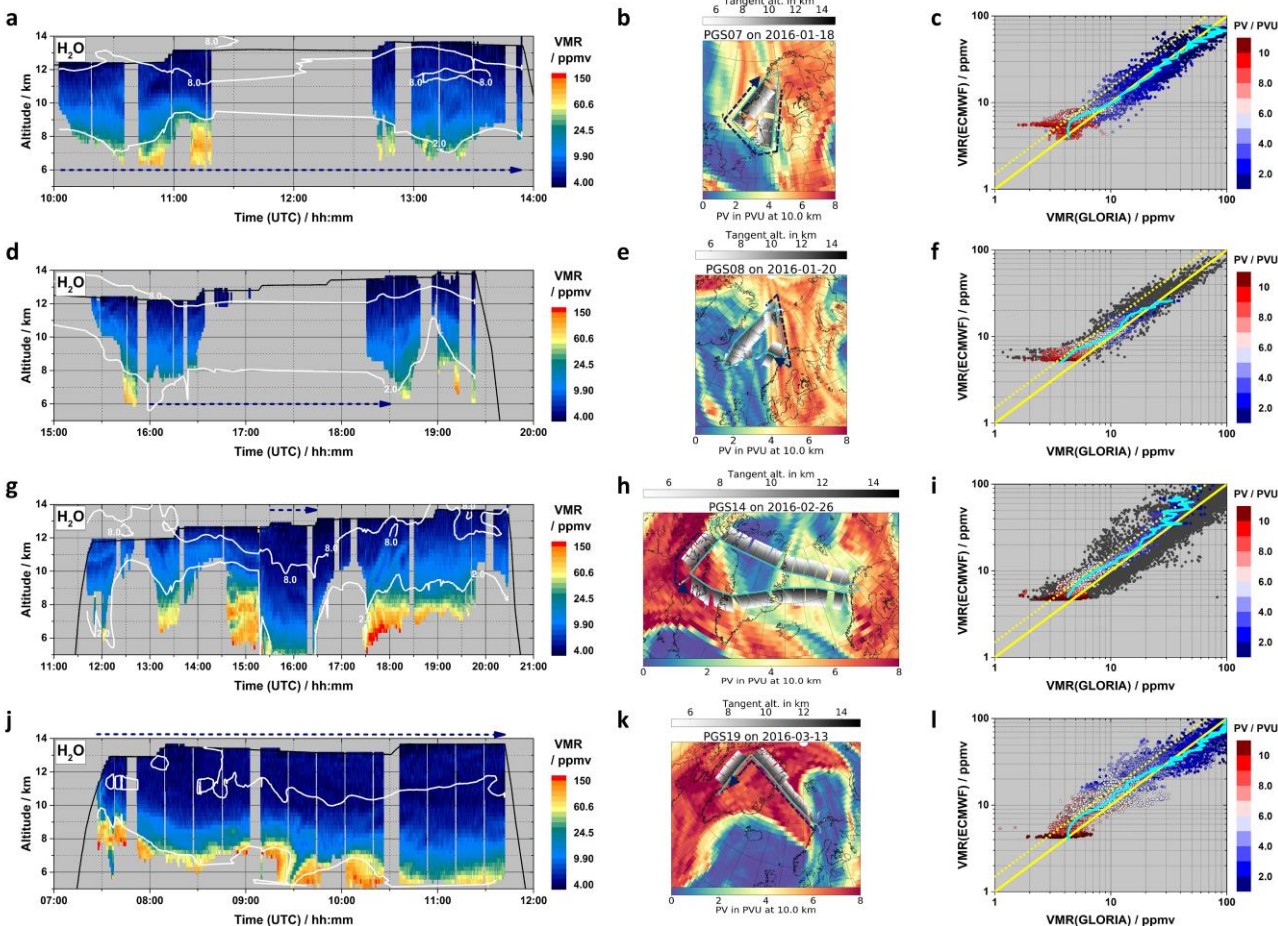

**Figure 2:** LMS moist bias quantification for selected flights from January to March 2016. GLORIA water vapour (left column), flight path and observation geolocations (middle column), and correlation IFS versus GLORIA (right column) for the flights on 18 January 2016, 20 January 2016, 26 February 2016, and 13 March 2016. For legend see Fig. 1.

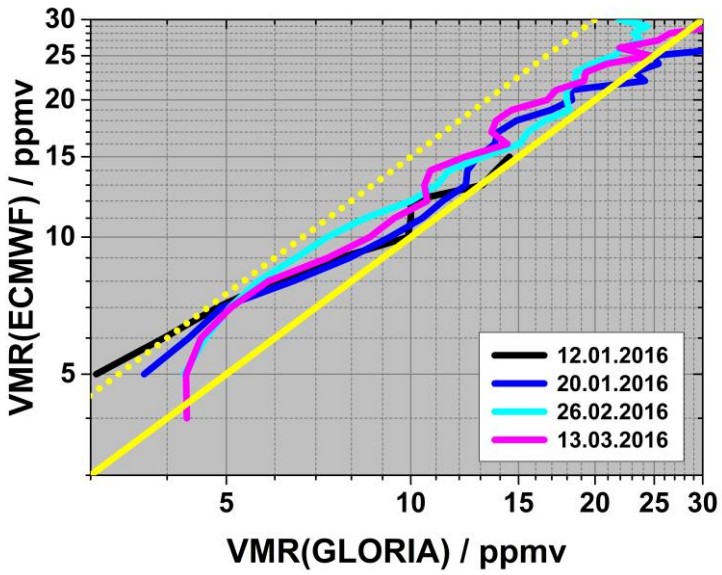

**Figure 3:** Overlay of mean correlations for flights from January to March 2016 (i.e. cyan lines in Fig. 1e and Fig. 2f,i,l).

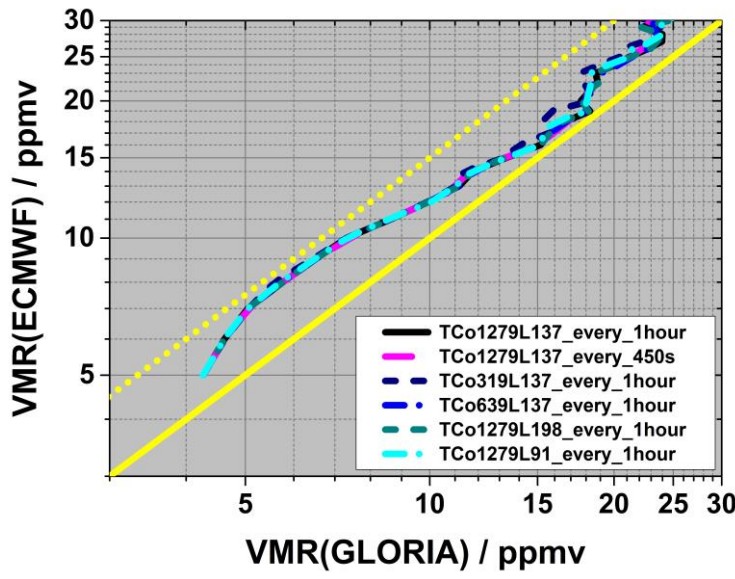


**Figure 4:** Low sensitivity of mean correlations between IFS sensitivity forecasts and GLORIA observations during the flight on 26 February 2016. IFS sensitivity forecasts include more frequent temporal output (450 sec (magenta) instead of 1 h (black)), lower/higher horizontal resolution ($T_{Co}319$ (dark blue); $T_{Co}639$ (blue); and $T_{Co}1279$ (black) all on cubic octahedral grid; instead of the operational $T_L1279$ on the linear grid) and higher/lower vertical resolution (L198 (dark cyan) and L91 (cyan) instead of L137 (black)).