# Peer review of "Technical note: Lowermost-stratospheric moist bias in ECMWF IFS model diagnosed from airborne GLORIA observations during winter/spring 2016"

_Atmospheric Chemistry and Physics, 2020_

## Referee Comment (RC1) · Anonymous Referee #1 · 6 Jun 2020

**Review of W. Woiwode et al. "Technical note: Lowermost-stratospheric moist bias in ECMWF IFS model diagnosed from airborne GLORIA observations during winter/spring 2016"**

 The technical note by Wolfgang Woiwode and coauthors addresses the quality of ECMWF Integrated Forecast System analysis and forecast data on upper troposphere and lowermost stratosphere humidity using remote sensing measurements onboard HALO aircraft. The validation of IFS analysis makes use of high-resolution observations by GLORIA thermal infrared spectrometer observations of water vapour collected during five HALO flights above Europe in January-March 2016 sampling polar sub-vortex air masses.  The model output is interpolated to the geolocations of GLORIA two-dimensional measurements and the intercomparison is performed in consideration of potential vorticity provided by IFS, which serves an indicator of the air mass origin and therefore its properties. The key outcome of the study is a confirmation of the moist bias in ECMWF LMS humidity already reported in the literature. The key figure pointed out throughout the article is the peak +50% wet bias compared to GLORIA.

The biases in UT/LS humidity data in ECMWF (re)analysis is certainly an important issue and even though the representability of the reported results may be questioned, the unique experimental setup and the thorough comparison approach render the study suitable for a technical note in ACP journal. The manuscript is well written and the graphical material is excellent.

My major concern on the manuscript is the oversold magnitude of wet bias in ECMWF and lacking discussion of the obtained results in the context of previous surveys.  I believe the scientific value of this study could be enhanced by addressing the following remarks.

**General remarks**

•   The introduction is too lengthy for a technical note and also misleading at times (see specific remarks). I would suggest to move most of it into discussion, where the results from the literature could be compared with the outcome of this study.

•   I was surprised to see $H_2O$ mixing ratios down to 2 ppmv or less reported by GLORIA in this region of the atmosphere. I am not aware of any published or unpublished $H_2O$ measurements revealing such dry values in the sub-polar UT/LS.  One might hypothesize that HALO has sampled an impressive dehydration event, however the anomalously dry values occur form one flight to another. The occasional dry artefacts in GLORIA can also be seen in Fig.10d of Johansson et al.  I wonder how  the individual and global difference profiles would look after a more rigorous filtering of GLORIA retrievals in the LMS.

•   As an IFS data user, I would be interested to see more statistical figures in the abstract and conclusions beside the peak value of 50%. Although the figures do give a good idea on the bias distribution as a function of mixing ratio and PV, I believe the concluding statements could be smoother about the IFS moist bias.

**Specific remarks**

p.2, ll.47-52. It makes one wonder why the most advanced ECMWF model with highest possible resolution still suffers from the moist and cold biases. Is this specific to the operational analysis/forecasts only or applies to any products including reanalysis?

p.2,ll.49-57. The paragraph mentions several potential reasons for the model wet biases, which should belong to the discussion. Could the reported findings support one or the other assumption?

p.2. l.65. What sort of gap is implied here?

---

## Referee Comment (RC2) · Anonymous Referee #2 · 11 Aug 2020

The paper by Woiwode et al. evaluates the lowermost stratospheric moist bias in ECMWF analyses and 12h forecasts using high spatial resolved water vapor mixing ratios from the airborne GLORIA observations. A systematic moist peaking at +50% at potential vorticity levels of 6-10 PVU is diagnosed. By model sensitivity experiments, the authors further show that the diagnosed moist bias is insensitive to model grid resolution for short-term forecasts and is present already in the initial conditions. The study is interesting and provides import information for model simulations with respect to lowermost stratospheric water vapor. The paper is well written and organized. I

recommend the publication of it subject to the technical corrections suggested below.

L51-52: 'too moist analysis' is unclear to me.

L56-57: How can 'small-scale stratospheric intrusions' affect LMS water vapor? Maybe you mean troposphere-to-stratosphere transport?

L84: Please check the value and unit for '0.0625 cm-1'.

L106-107: There seems to be a grammatical problem for the phrase 'from a 12 h deterministic forecasts in between the analysis cycles'.

---

## Author Comment (AC1) · 18 Sep 2020

**Author response to comments by Anonymous Referee #1 to: "Technical note: Lowermost-stratospheric moist bias in ECMWF IFS model diagnosed from airborne GLORIA observations during winter/spring 2016"**

Atmos. Chem. Phys. Discuss., https://doi.org/10.5194/acp-2020-367, in review, 2020
Woiwode et al.

We thank Referee #1 for his/her time and valuable comments to improve the manuscript. In the following, we provide the original referee comments (italic letters), followed by our responses.

*The technical note by Wolfgang Woiwode and coauthors addresses the quality of ECMWF Integrated Forecast System analysis and forecast data on upper troposphere and lowermost stratosphere humidity using remote sensing measurements onboard HALO aircraft. The validation of IFS analysis makes use of high-resolution observations by GLORIA thermal infrared spectrometer observations of water vapour collected during five HALO flights above Europe in January-March 2016 sampling polar sub-vortex air masses. The model output is interpolated to the geolocations of GLORIA two-dimensional measurements and the intercomparison is performed in consideration of potential vorticity provided by IFS, which serves an indicator of the air mass origin and therefore its properties. The key outcome of the study is a confirmation of the moist bias in ECMWF LMS humidity already reported in the literature. The key figure pointed out throughout the article is the peak +50% wet bias compared to GLORIA.*

*The biases in UT/LS humidity data in ECMWF (re)analysis is certainly an important issue and even though the representability of the reported results may be questioned, the unique experimental setup and the thorough comparison approach render the study suitable for a technical note in ACP journal. The manuscript is well written and the graphical material is excellent.*

We appreciate the encouraging statement by Referee #1.

*My major concern on the manuscript is the oversold magnitude of wet bias in ECMWF and lacking discussion of the obtained results in the context of previous surveys. I believe the scientific value of this study could be enhanced by addressing the following remarks.*

We thoroughly revised the manuscript under consideration of the remarks by the Referee. We included further references and provide more context to previous work. However, we consider the wet bias problem to be important, since radiative forcing is known to be sensitive to LMS water vapour mixing ratios, ECMWF IFS data products are widely used, and the problem is still not solved. We also note that specific humidity observations are not assimilated in any ECMWF products above the tropopause. Therefore, the presence of a moist bias in the stratosphere is not surprising. Moreover, the moist bias magnitude in the lowermost stratosphere in our study is consistent with previous comparisons of ECMWF analysis against independent specific humidity observations (Dyroff et al. 2015; Shepherd et al., 2018), which are discussed in our study. In the same context, we now included further references to Kaufmann et al. (2018) and Kuntz et al. (2014).

**General remarks**

- *The introduction is too lengthy for a technical note and also misleading at times (see specific remarks). I would suggest to move most of it into discussion, where the results from the literature could be compared with the outcome of this study.*

Following the suggestion by the Referee, we moved parts of the introduction into Section 3 "Results and Discussion" and updated the discussion accordingly. Since this Section would become relatively long, we subdivided it into Section 3 "Results" and Section 4 "Discussion".

- *I was surprised to see H2O mixing ratios down to 2 ppmv or less reported by GLORIA in this region of the atmosphere. I am not aware of any published or unpublished H2O measurements revealing such dry values in the sub-polar UT/LS. One might hypothesize that HALO has sampled an impressive dehydration event, however the anomalously dry values occur form one flight to another. The occasional dry artefacts in GLORIA can also be seen in Fig.10d of Johansson et al. I wonder how the individual and global difference profiles would look after a more rigorous filtering of GLORIA retrievals in the LMS.*

We thank the Referee for pointing out this issue, which clearly requires more explanation. As discussed by Johansson et al. (2018), the indicated GLORIA errors are combined random and systematic 1σ errors. Therefore, under consideration of a conservative 1σ error of (10-)20%, the 3σ range at a "real" water mixing ratio of 4.0 ppmv ranges from 1.6 ppmv to 6.4 ppmv, which is well consistent with the scattering of the individual data points in the correlation plots (see Figs. 1e and 2c,f,i,l). Therefore, the dry values in the correlations are not indicative of an extreme dehydration event, but are caused by scattering in the GLORIA data due to their uncertainties.

Note, due to the large number of data points in the correlations, such strongly scattered data points appear somewhat overrepresented when compared to less scattering data points, which are mostly hidden behind other data points. The scattering towards dry values is furthermore emphasised by the logarithmic scale of the axes.

As discussed in Section 2.1, the errors in the GLORIA data are expected to cancel out mostly when the data are analysed as ensemble (e.g. in correlation analyses). Therefore, we use the mean correlation of the data point ensembles of entire flights (cyan lines in the correlations) instead of single GLORIA data points (or individual GLORIA profiles) to quantify the wet bias.

As correctly pointed out by the Referee, relatively dry data points are also seen in Figure 10d of Johansson et al. (2018). These values are also explained by scattering in the GLORIA data due to uncertainties. Also here, the logarithmic scale emphasises "dry" values. Furthermore, the relatively dry section with values down to ~3 ppmv around 11:00 in the same figure might be explained by an enhanced systematic error component during this short part of the flight and/or the fact that GLORIA (pointing to the right) observed e.g. a dry filament beside the flight path which was not present at the flight path itself (i.e. where the in situ measurements were done). However, as shown in Johansson et al. (2018) Table 2, GLORIA showed good overall agreement with FISH during the same flight.

For clarification, we now mention in Section 2 that the GLORIA data are combined random and systematic 1σ errors and discuss the consequences in Section 3 in context of the correlation plots.

- *As an IFS data user, I would be interested to see more statistical figures in the abstract and conclusions beside the peak value of 50%. Although the figures do give a good idea on the bias distribution as a function of mixing ratio and PV, I believe the concluding statements could be smoother about the IFS moist bias.*

To quantify the moist bias, we use mean correlations from large data point ensembles of entire flights. From our point of view, statistical numbers on the correlations would mainly reflect the scattering due to the uncertainties in the GLORIA data. Therefore, we think that it is difficult to extract further information from statistical quantifiers here. From our point of view, the mean correlation plots of the different flights are conclusive and draw a coherent picture of the moist bias and its evolution.

However, possibly we did not correctly seize the suggestion by the Referee, and we are open to further suggestions.

***Specific remarks***

*p.2, ll.47-52. It makes one wonder why the most advanced ECMWF model with highest possible resolution still suffers from the moist and cold biases. Is this specific to the operational analysis/forecasts only or applies to any products including reanalysis?*

This applies to all ECMWF products. This is because no specific humidity observations are assimilated above the hygropause (which is used as a proxy for the tropopause) in any ECMWF analyses/reanalyses (i.e. humidity increments are turned off there). Above, specific humidity is determined essentially by the model. However, temperature and wind are still assimilated above the hygropause (i.e. wind and temperature increments remain turned on there) and may influence humidity. The resolution of reanalyses is lower than that of the operational analysis. Therefore, we expect the bias to be larger in reanalysis than in the high resolution analysis.

*p.2,ll.49-57. The paragraph mentions several potential reasons for the model wet biases, which should belong to the discussion. Could the reported findings support one or the other assumption?*

As suggested by the Referee, we moved this part to the discussion and comment on how our results compare with the other studies. Our results confirm the bias to be present in the forecast initialisation (i.e., the analysis) and the vertical structure (i.e., decreasing bias towards higher altitudes), and further insights by our study are discussed. We slightly revised our wording in the abstract: our results suggest a model bias together with the lack of water vapour observations suitable for assimilation by the model above the tropopause to cause the bias.

*p.2. l.65. What sort of gap is implied here?*

Here, we mean the "gap" in terms of spatial coverage and resolution. We rephrased the sentence for clarification.

**References**

Kaufmann, S., Voigt, C., Heller, R., Jurkat-Witschas, T., Krämer, M., Rolf, C., Zöger, M., Giez, A., Buchholz, B., Ebert, V., Thornberry, T., and Schumann, U.: Intercomparison of midlatitude tropospheric and lower-stratospheric water vapor measurements and comparison to ECMWF humidity data, Atmos. Chem. Phys., 18, 16729–16745, https://doi.org/10.5194/acp-18-16729-2018, 2018

Kunz, A., Spelten, N., Konopka, P., Müller, R., Forbes, R. M., and Wernli, H.: Comparison of Fast In situ Stratospheric Hygrometer (FISH) measurements of water vapor in the upper troposphere and lower stratosphere (UTLS) with ECMWF (re)analysis data, Atmos. Chem. Phys., 14, 10803–10822, https://doi.org/10.5194/acp-14-10803-2014, 2014

---

## Author Comment (AC2) · 18 Sep 2020

**Author response to comments by Anonymous Referee #2 to: "Technical note: Lowermost-stratospheric moist bias in ECMWF IFS model diagnosed from airborne GLORIA observations during winter/spring 2016"**

Atmos. Chem. Phys. Discuss., https://doi.org/10.5194/acp-2020-367, in review, 2020
Woiwode et al.

We thank Referee #2 for his/her time and valuable comments to improve the manuscript. In the following, we provide the original referee comments (italic letters), followed by our responses.

*The paper by Woiwode et al. evaluates the lowermost stratospheric moist bias in ECMWF analyses and 12h forecasts using high spatial resolved water vapor mixing ratios from the airborne GLORIA observations. A systematic moist peaking at +50% at potential vorticity levels of 6-10 PVU is diagnosed. By model sensitivity experiments, the authors further show that the diagnosed moist bias is insensitive to model grid resolution for short-term forecasts and is present already in the initial conditions. The study is interesting and provides import information for model simulations with respect to lowermost stratospheric water vapor. The paper is well written and organized. I recommend the publication of it subject to the technical corrections suggested below.*

We appreciate the positive statement by Referee #2.

*L51-52: 'too moist analysis' is unclear to me.*

We agree that clarification is required and rephrased to "… from too moist conditions in the analysis …"

*L56-57: How can 'small-scale stratospheric intrusions' affect LMS water vapor? Maybe you mean troposphere-to-stratosphere transport?*

We thank Referee #2 for pointing out this mistake. We rephrased to "… small scale tropospheric intrusions …"

*L84: Please check the value and unit for '0.0625 cm-1'.*

We verified this value, it corresponds to the unapodized spectral sampling in wave numbers, which is associated with an optical path difference of the interferometer of 8 cm in the "chemistry mode" (Friedl-Vallon et al., 2014). Note, application of the Norton-Beer "strong" apodization in level 1 processing results in an effective spectral resolution of 0.121 cm$^{-1}$ in this specific measurement mode.

*L106-107: There seems to be a grammatical problem for the phrase 'from a 12 h deterministic forecasts in between the analysis cycles'.*

We corrected the phrase by removing "a".

---

## Author Comment (AC3) · 18 Sep 2020

We thank both Referees for their time and valuable comments which helped us to improve the manuscript.

In the supplement, we summarise our responses to both Referees. Furthermore, we discuss additional technical corrections and updates by the authors.

Sincerely, Wolfgang Woiwode, on behalf of all co-authors

[Figure]

Please also note the supplement to this comment:
https://acp.copernicus.org/preprints/acp-2020-367/acp-2020-367-AC3-supplement.pdf

———————————————————

---

## Author Response (AR3)

**Author response to comments by Anonymous Referee #1, submitted on 03 Oct 2020, to: "Technical note: Lowermost-stratospheric moist bias in ECMWF IFS model diagnosed from airborne GLORIA observations during winter/spring 2016"**

Atmos. Chem. Phys. Discuss., https://doi.org/10.5194/acp-2020-367, in review, 2020
Woiwode et al.

We thank Referee #1 for his/her time and further valuable comments to improve the manuscript. In the following, we provide the original referee comment (italic letters), followed by our responses.

*The authors have addressed my concerns and the article is in better shape now.*

We thank Referee #1 for this encouraging statement.

*I however still have a slight concern regarding the way how the moist bias is presented in the manuscript. Whenever possible the bias is reported as a peak value that approaches or exceeds +50 %. This value is encountered in the text 9 times for every occasion.*

We removed the value of +50 % at several occasions. In section 3, we now provide more detailed values of the moist bias with respect to potential vorticity and discuss these values accordingly.

*I still believe it would be useful to provide other statistical values in addition to the peak one, which seems to be independent of PVU and flight date.*

Following the suggestion by the Referee, we included a more detailed statistical analysis. We now calculate the mean bias and the standard deviation of the individual data points at selected, rounded levels of potential vorticity. Note that slightly different values are diagnosed here when compared to the mean correlations, since the mean correlations are a function of volume mixing ratio and not potential vorticity. Using this approach, we find maximum values of +70 % (data points at a rounded level of 10 PVU on 12 January 2016) to +35 % (data points at a rounded level of 7 PVU on 13 March 2016). However, the overall conclusions remain the same. We revised section 3, the abstract and the conclusions accordingly. We thank Referee #1 for encouraging a more detailed analysis here.

**Further modification by the authors**

We revisited the in situ comparisons by Johansson et al. (2018) in context currently ongoing refinement of the GLORIA data processing and noticed differences in the results for the phase from December 2015 – January 2016 (FISH data systematically higher than GLORIA on average) and February 2016 – March 2016 (differences between FISH and GLORIA approximately cancel out on average). 
[revised manuscript text omitted]